# COVID-19 and Seasonal Influenza Vaccination: Cross-Protection, Co-Administration, Combination Vaccines, and Hesitancy

**DOI:** 10.3390/ph15030322

**Published:** 2022-03-08

**Authors:** Alexander Domnich, Andrea Orsi, Carlo-Simone Trombetta, Giulia Guarona, Donatella Panatto, Giancarlo Icardi

**Affiliations:** 1Hygiene Unit, San Martino Policlinico Hospital-IRCCS for Oncology and Neurosciences, 16132 Genoa, Italy; andrea.orsi@unige.it (A.O.); giuly.guarons@outlook.it (G.G.); icardi@unige.it (G.I.); 2Department of Health Sciences (DISSAL), University of Genoa, 16132 Genoa, Italy; s5285039@studenti.unige.it (C.-S.T.); panatto@unige.it (D.P.)

**Keywords:** influenza, COVID-19, vaccination, vaccine co-administration, combination vaccines

## Abstract

SARS-CoV-2 and influenza are the main respiratory viruses for which effective vaccines are currently available. Strategies in which COVID-19 and influenza vaccines are administered simultaneously or combined into a single preparation are advantageous and may increase vaccination uptake. Here, we comprehensively review the available evidence on COVID-19/influenza vaccine co-administration and combination vaccine candidates from the standpoints of safety, immunogenicity, efficacy, policy and public acceptance. While several observational studies have shown that the trained immunity induced by influenza vaccines can protect against some COVID-19-related endpoints, it is not yet understood whether co-administration or combination vaccines can exert additive effects on relevant outcomes. In randomized controlled trials, co-administration has proved safe, with a reactogenicity profile similar to that of either vaccine administered alone. From the immunogenicity standpoint, the immune response towards four influenza strains and the SARS-CoV-2 spike protein in co-administration groups is generally non-inferior to that seen in groups receiving either vaccine alone. Several public health authorities have advocated co-administration. Different combination vaccine candidates are in (pre)-clinical development. The hesitancy towards vaccine co-administration or combination vaccines is a multifaceted phenomenon and may be higher than the acceptance of either vaccine administered separately. Public health implications are discussed.

## 1. Introduction

Vaccination against COVID-19 is a cornerstone public health intervention to tackle the ongoing pandemic [1]. Analogously, seasonal influenza vaccination (SIV) is considered one of the most effective means of reducing the burden of disease, which in the pre-COVID-19 era caused on average 250,000–500,000 deaths worldwide each year [2,3]. Although circulation of the influenza virus has diminished drastically since 2020, the ongoing 2021/22 northern hemisphere winter season is characterized by the circulation of both SARS-CoV-2 and influenza [4]. Moreover, in the 2021/22 season, there is an overlap between the administration of booster doses of COVID-19 vaccines and the SIV campaign [5,6].

Very little is yet known about the interaction between SARS-CoV-2 and influenza viruses. Dadashi et al. [7] estimated a pooled prevalence of SARS-CoV-2 and influenza co-infection of 0.8% (95% confidence interval (CI): 0.4–1.3%) with marked regional heterogeneity, ranging from 0.4% in the Americas to 4.5% in Asia. A large study conducted in England [8] reported that, while individuals positive for influenza had 58% (95%: 44–69%) lower odds of also testing positive for SARS-CoV-2, co-infected individuals had worse clinical outcomes. Specifically, co-infected patients were approximately twice as likely to die (odds ratio (OR) 2.27; 95% CI: 1.23–4.19) as subjects positive only for SARS-CoV-2 [8]. Some experimental evidence has provided useful insights into these poorer outcomes in co-infected patients. Indeed, it has been observed [9] that prior infection with type A influenza virus promotes SARS-CoV-2 entry and infectiousness in both cell and animal models, probably owing to the ability of the former to increase the expression of angiotensin-converting enzyme 2 (ACE2). Interestingly, respiratory syncytial virus (RSV), parainfluenza virus and human rhinovirus 3 had no effect on SARS-CoV-2 infection [9]. Finally, a robust mathematical model by Domenech de Cellès et al. [10] indicated that influenza virus could have facilitated the spread of SARS-CoV-2 during the early 2020 COVID-19 pandemic phase in Europe.

As mentioned earlier, the available SIV and COVID-19 vaccines are essential to controlling both infections. However, simply having effective vaccines does not necessarily mean achieving public health goals. Indeed, coverage of both COVID-19 [11] and SIV [12,13] vaccines is still suboptimal, or even low, in several jurisdictions and target populations. Interventions to increase immunization coverage rates can be conceptually divided into three broad categories: (i) interventions to increase community demand, (ii) interventions to enhance access, and (iii) provider-based interventions [14]. The co-administration of vaccines (i.e., when two or more vaccines are administered during the same visit) or the use of combination/multivalent vaccines (i.e., vaccines containing two or more antigens in the same preparation) may enhance access to immunization in several ways. Indeed, these strategies have several potential benefits, including improved patient convenience and compliance, simplified immunization schedules, fewer missed opportunities to vaccinate, reduced costs, and logistical advantages [15,16,17,18].

The aim of this study was to provide a state-of-the-art overview of current research into the safety, immunogenicity and efficacy of COVID-19 and SIV vaccine co-administration, mechanisms beyond possible immunological inference, combined COVID-19/SIV candidates and available public health policies and recommendations.

## 2. Non-Specific Effects of Seasonal Influenza Vaccination on COVID-19-Related Outcomes

### 2.1. Real-World Evidence

Some early ecological studies [19,20], which are useful for hypothesis generation [21], found a negative correlation between SIV coverage rates and COVID-19-related outcomes. For instance, in Italy, which was the first western country where SARS-CoV-2 spread widely, a negative correlation (*r*  =  −0.59, *p*  =  0.005) between regional SIV coverage rates in the elderly and COVID-19 deaths was demonstrated [19]. However, subsequent cohort studies on this non-specific effect of SIV yielded contrasting results: some [22,23,24] found a protective effect, while others did not find any significant association [25,26]. To summarize the body of available observational studies, Wang et al. [27] conducted a systematic review and meta-analysis of this non-specific association. In their random-effects model, a significant reduction in laboratory-confirmed cases of SARS-CoV-2 was found in subjects immunized with SIV, with a pooled (*n* = 9 studies) OR of 0.86 (95% CI: 0.79–0.94). On the other hand, the association between SIV and some other COVID-19-related outcomes, such as hospitalization (OR 0.74; 95% CI: 0.51–1.06), admission to intensive care units (OR 0.63; 95% CI: 0.22–1.81) or mortality (OR 0.89; 95% CI: 0.73–1.09) was not statistically significant [27].

### 2.2. Underlying Immunological Mechanisms

Different immunological mechanisms beyond the observed non-specific heterologous effects of SIV on COVID-19-related clinical endpoints, have been proposed. These mechanisms may involve either innate (the so-called “trained immunity”) or adaptive (bystander activation and cross-reactivity) compartments of the immune system [28]. Bystander activation refers to a type of heterologous response which is exerted by adjacent, but not relevant, T cells with different specificity. These heterologous T cells are probably activated by cytokines as a result of the activation of cells during the classical response [29]. By contrast, the cross-reactivity theory holds that T cells involved in the classical adaptive immune response may cross-react with an antigen presenting some degree of amino acid similarity [28]. Finally, the trained immunity hypothesis postulates that the innate immune cells may be primed upon encountering exogenous or endogenous insults, causing long-term metabolic and epigenetic reprogramming of these cells and leading to an enhanced response to a second challenge [30,31,32].

The available experimental data on influenza virus- and/or SIV-induced cross-reactive or even cross-protective antibodies against SARS-CoV-2 are controversial. For instance, with regard to T and B cell reactivity, Reche [33] concluded that influenza viruses do not have epitopes that cross-react with SARS-CoV-2. Murugavelu et al. [34] tested polyclonal sera obtained from SARS-CoV-2-positive subjects with high anti-spike neutralizing antibody titers and found some degree of cross-reactivity with influenza virus hemagglutinin in both enzyme-linked immunosorbent (ELISA) and Western blot assays. However, a subsequent analysis demonstrated that these hemagglutinin cross-reactive binding antibodies were not neutralizing. More recently, Almazán et al. [35] investigated the role of the small NGVEGF peptide—which is identical, or very similar, to a peptide found in most contemporary A(H1N1)pdm09 strains—in inducing cross-reactive antibodies. This peptide is present in the most critical part (N481–F486) of the receptor binding domain (RBD) of the SARS-CoV-2 spike protein, which interacts with the ACE2 receptor, while in influenza A(H1N1)pdm09 strains, the NGVEGF/NGVKGF peptide is located in an immunodominant region of the neuraminidase. Approximately two thirds of blood donors (*n* = 328) had detectable levels of antibodies to this peptide. Immunization with a quadrivalent egg-based influenza vaccine (QIVe) enhanced the anti-SARS-CoV-2 response: subjects with no recent influenza infection had low binding inhibitory activity (average of 32.7%), which was enhanced by QIVe administration (average of 55%) and further enhanced by the BNT162b2 (Comirnaty; Pfizer Inc., New York, NY, USA and BioNTech, Mainz, Germany) vaccine (average of 94%). The NGVEGF peptides also activated CD8+ cells in 20% of donors. Finally, the authors identified 11 additional CD8+ cell peptides that potentially cross-reacted with both SARS-CoV-2 and influenza viruses; depending on the type of human leukocyte antigen (HLA), these peptides may protect against SARS-CoV-2 in about 40–71% of individuals [35].

The bystander activation mechanism has been partially proven by Pallikkuth et al. [36]. Specifically, in their cohort of healthcare workers, A(H1N1) antigen-specific CD4+ cells were present in 92% and 76% of SARS-CoV-2-positive and -negative subjects, respectively. The A(H1N1) CD4+ response also showed a strong positive correlation with SARS-CoV-2-specific CD4+ cells [36].

The trained immunity hypothesis has recently drawn particular attention. It was first documented in the case of BCG (Bacillus Calmette–Guérin) vaccine, and then measles, oral polio and, more recently, SIV [28,29]. Experimental confirmation of SIV-induced trained immunity against SARS-CoV-2 was recently obtained in a Dutch study [37]. Following the demonstration of a 37–49% relative risk reduction of SARS-CoV-2 infection among healthcare workers vaccinated with QIVe (compared with non-vaccinated subjects), the authors investigated the biological plausibility of this observation in a well-established in vitro model. Specifically, following the stimulation of peripheral blood mononuclear cells with QIVe and BCG, an increase in the production of cytokines was observed. Re-stimulation of these cells with a heat-inactivated SARS-CoV-2 strain induced a higher production of interleukin (IL)-1 receptor antagonist (IL-1RA), while the production of pro-inflammatory IL-1*β* and IL-6 was reduced [37]. An Italian study [38] conducted among healthcare workers (*n* = 710) who received 2 doses of the BNT162b2 vaccine found that, in subjects previously vaccinated with the quadrivalent cell culture-based influenza vaccine (QIVc; Flucelvax Tetra, Seqirus Netherlands B.V., Amsterdam, The Netherlands) plus pneumococcal vaccines or with QIVc alone, microneutralization titers against SARS-CoV-2 were 58% (*p* = 0.01) and 42% (*p* = 0.07) higher, respectively, than in subjects who did not receive any vaccine. By contrast, no significant differences were found for the anti-spike and interferon-*γ* responses [38]. Another Italian study [39] compared the immune response between healthcare workers who received 2 doses of the BNT162b2 vaccine and those who also received a dose of QIVc. In the overall cohort, the anti-spike RBD antibodies were on average 37% higher (2047 vs. 1494; *p* = 0.0039) in the former group. This difference was mainly driven by subjects aged ≥35 years. However, antibody waning was substantially greater in subjects who had received both QIVc and BNT162b2 vaccines, and 2 months after the second COVID-19 dose, antibody titers were similar to those seen in participants immunized with BNT162b2 only [39].

## 3. Safety, Immunogenicity and Efficacy of COVID-19 and Influenza Vaccine Co-Administration

As of February 2022, three randomized controlled trials (RCTs) [40,41,42] on COVID-19 and SIV vaccine co-administration are available. In a phase IV trial conducted in the United Kingdom (UK) [40], adults were randomized (1:1; *n* = 679) to receive either a second dose of ChAdOx1 (Vaxzevria, AstraZeneca, Cambridge, UK) or BNT162b2 vaccines, together with an age-appropriate SIV (QIVc, recombinant quadrivalent influenza vaccine (QIVr; Supemtek, Sanofi Pasteur, Lyon, France) for subjects aged 18–64 years and MF59-adjuvanted trivalent influenza vaccine (aTIV; Fluad, Seqirus) for those aged ≥65 years) or ChAdOx1/BNT162b2 together with placebo. Three weeks later, those who had received placebo received SIV, and vice versa. Toback et al. [41] reported the results of a phase III efficacy RCT, in which a subset of adults was randomized (1:1; *n* = 431) to receive the first dose of NVX-CoV2373 (Nuvaxovid, Novavax CZ a.s., Jevany, Czech Republic) plus SIV (QIVc and aTIV for subjects aged 18–64 and ≥65 years, respectively) or SIV alone. Finally, the interim results of a phase II RCT have recently been reported [42]; in this trial, elderly individuals (≥65 years) were randomized (1:1:1; *n* = 431) to receive a second dose of mRNA-1273 (Spikevax, Moderna, Cambridge, MA, USA) plus a high-dose quadrivalent influenza vaccine (hdQIV; Fluzone High-Dose Quadrivalent, Sanofi Pasteur, Lyon, France), a dose of hdQIV alone or a second dose of mRNA-1273 alone. In all three RCTs [41,42], vaccines were co-administered in opposite arms in each subject.

All three RCTs [41,42] reported no major safety concerns regarding COVID-19 + SIV co-administration. Specifically, as reported in Table 1, the overall rate of solicited local (especially pain in the injection site) and systemic adverse events was similar between subjects who received COVID-19 + SIV and those who received the COVID-19 vaccine alone. The adverse events reported were mostly mild-to-moderate and self-limiting. A similar picture was seen with regard to unsolicited adverse events.

It has generally been found [40,41,42] that the humoral IgG response measured by means of the hemagglutination inhibition (HAI) assay approximately 3 weeks after immunization towards any SIV strain is preserved in COVID-19 + SIV co-administration groups. Indeed, apart from the statistical significance, geometric mean ratios of all pairwise comparisons have proved to be >0.67, which is the non-inferiority margin (Table 2). Lazarus et al. [40] did not observe any significant difference in geometric mean titers (GMTs) between most co-administration groups (ChAdOx1 + QIVc, ChAdOx1 + QIVr, ChAdOx1 + aTIV, BNT162b2 + QIVc or BNT162b2 + aTIV) and groups receiving SIVs alone. The only exception was the immune response to A(H1N1)pdm09, B/Victoria and B/Yamagata; in these cases, GMTs were 20–38% higher in the BNT162b2 + QIVr group than in individuals who received QIVr only. The IgG response to A(H3N2) was similar. Toback et al. [41] did not find any significant difference in HAI titers between COVID-19 + SIV and placebo + SIV groups, regardless of the strain or vaccine (QIVc or aTIV). Analogously, the humoral immune response towards all four SIV strains was similar in individuals who received mRNA-1273 + hdQIV or hdQIV alone [42].

Similar results were reported with regard to anti-spike neutralizing antibody titers (Table 3). Lazarus et al. [40] and Izikson et al. [42] did not find any significant differences between the co-administration groups and individuals who received COVID-19 vaccines alone. By contrast, some immunological inference was reported in the RCT by Toback et al. [41] After adjustment for baseline titers, age and the treatment arm, the geometric mean ratio of the anti-spike humoral response in NVX-CoV2373 + QIVc/aTIV versus NVX-CoV2373 alone was 0.57 (95% CI: 0.47–0.70). On the other hand, this decrease did not seem to translate into a corresponding decrease in vaccine efficacy. For instance, absolute vaccine efficacy against laboratory-confirmed symptomatic COVID-19 in adults (18–64 years) was 87.5% (95% CI: −0.2–98.4%) and 89.8% (95% CI: 79.7–95.5%) in the influenza sub-study and main study, respectively [41].

Finally, a phase III RCT on the safety and immunogenicity of the co-administration of Ad26.COV2.S (Janssen: Pharmaceutical Companies of Johnson & Johnson, New Brunswick, NJ, USA) and QIVe vaccines in adults is ongoing, and is expected to end on 31 August 2022 [43].

## 4. The Current Position of Some Authorities on COVID-19 and Influenza Vaccine Co-Administration

Following the examination of the available evidence, the World Health Organization (WHO) provided regularly updated interim guidelines [44], on the co-administration of COVID and SIV vaccines. While recognizing the limits of the currently available evidence, the WHO suggests that the co-administration of any inactivated SIV and any dose of any approved COVID-19 vaccine is a feasible option, as it is logistically easier and can increase coverage. In order to reduce any perceived risk, the WHO advises using contralateral arms for vaccination [44]. The interim guidelines issued by the Centers for Disease Control and Prevention (CDC) [45] state that COVID-19 vaccines may be administered without regard to the timing of other vaccines, including SIV. Vaccines should be administered at different injection sites [45]. Similar recommendations have been provided by Public Health Authorities in other countries, including Italy [46], France [47], Germany [48], Spain [49], Finland [50], the UK [51], Russia [52] and Australia [53]. However, in some countries, such as Australia, earlier versions of these recommendations, which were issued before clinical data on vaccine co-administration became available, advised a precautionary time window of 7–14 days between COVID-19 and SIV administration. This strategy may have had some negative impact on the uptake of SIV [54].

## 5. COVID-19/Influenza Combination Vaccine Candidates

The immunogenicity, efficacy and safety of a combination vaccine may be affected by immunological interactions, which may be not only due to the combination of antigens, but also other vaccine components, such as adjuvants and other excipients. The clinical development of combined vaccines is demanding for several reasons. First, a combined vaccine is called upon to replace the corresponding single vaccines, and placebo-controlled trials may be seen as unethical [55]. This is the case in both COVID-19 and SIV, since some population groups, such as the elderly and at-risk adults, are the primary targets for both vaccines [56,57]. Moreover, clinical research is further complicated by the lack of a recognized correlate of protection for COVID-19 vaccines [58]. Finally, from the safety perspective, it may be not clear which vaccine component or manufacturing step is involved in causing a particular adverse reaction [55].

A phase I/II RCT on a combined COVID-19/SIV vaccine was undertaken in Autumn 2021 by Novavax and is expected to end in March 2022 [59]. In this observer-blinded study, a total of 642 adults aged 50–70 years, subdivided into 16 arms, will be randomized to receive two doses of different formulations of the combined vaccine, SIV alone or COVID-19 vaccine alone. SIV consists of a quadrivalent nanoparticle formulation adjuvanted with Matrix-M (qNIV; Nanoflu, Novavax), while the COVID-19 vaccine is Matrix-M-adjuvanted NVX-CoV2373. The SARS-CoV-2 spike protein and influenza hemagglutinins are produced by means of recombinant technology using a baculovirus expression system in an insect cell line derived from Sf9 cells of the *Spodoptera frugiperda* species. Immunogenicity towards influenza viruses will be measured by means of HAI and microneutralization assays, while the immune response induced by the SARS-CoV-2 spike protein will be quantified in ELISA and microneutralization assays [59]. Numerous experimental demonstrations of the potent adjuvant properties and good safety profile of Matrix-M, and successful clinical development of both qNIV and NVX-CoV2373, are the main premises for this combination vaccine, as described below.

Matrix-M is a saponin-based adjuvant derived from fractionated *Quillaja saponaria* Molina extract, phosphatidylcholine and cholesterol formulated into cage-like structures of approximately 40 nm in diameter [60,61]. Its adjuvant activity enhances both humoral and cellular responses: high- and long-lasting levels of broadly reacting antibodies are supported by a balanced Th1/Th2 response, including multifunctional and cytotoxic T cells. Matrix-M promotes effective cellular drainage to local lymph nodes, creating an immunocompetent environment of activated T, B, natural killer, neutrophil, monocyte, and dendritic cells [60,61,62,63].

The Matrix-M-adjuvanted NVX-CoV2373 vaccine has recently gained conditional approval [64] and has proved highly efficacious in preventing COVID-19 [41,65,66,67]. In a pivotal (*n* = 15,187) phase III RCT, overall vaccine efficacy was 89.7% (95% CI: 80.2–94.6%) against any symptomatic laboratory-confirmed COVID-19, and similar results were obtained in non-elderly (89.8%; 95% CI: 79.7–95.5%) and elderly (88.9%; 95% CI: 20.2–99.7%) subjects [65]. Another large (*n* = 29,949) pivotal RCT showed overall vaccine efficacy of 90.4% (95% CI: 82.9–94.6%), while efficacy against moderate-to-severe COVID-19 was 100% (95% CI: 87.0–100%) [66]. Finally, Shinde et al. [67] reported 51.0% (95% CI: −0.6–76.2%) efficacy against the Beta variant of concern (the vaccine was based on a prototype Wuhan-Hu-1 sequence).

Analogously, potent humoral and cellular immune responses induced by qNIV or its earlier trivalent formulation (tNIV) have been observed in a number of comparative clinical studies [68,69,70]. In comparison with the high-dose (60 μg) trivalent influenza vaccine (hdTIV; Fluzone High Dose, Sanofi Pasteur), the 60 μg formulation of tNIV induced 28–64% higher HAI titers against both vaccine-like strains and four drifted A(H3N2) strains, while HAI titers against homologous A(H1N1)pdm09 and B/Victoria strains were similar [68]. Another RCT [69] compared the immunogenicity produced by qNIV, hdTIV and rQIV. The HAI response against homologous and heterologous A(H3N2) strains was higher for qNIV than for hdTIV, but the response against A(H1N1)pdm09 and B/Victoria vaccine components was similar. Conversely, the magnitude of the immune response against all strains tested was similar between qNIV and rQIV. A qNIV formulation with a higher Matrix-M content (75 µg) produced considerably higher GMRs of hemagglutinin-specific polyfunctional CD4+ cells than both hdTIV and rQIV. In a more recently published phase III RCT [70], qNIV administered to the elderly induced qualitatively and quantitatively enhanced humoral and cellular immune responses in comparison with QIVe.

The above-described phase I/II RCT on the safety and immunogenicity of the qNIV/NVX-CoV2373 combination vaccine [59] is supported by the successful results of a pre-clinical study [71]. In their ferret and hamster models, Massare et al. [71] observed high HAI and neutralizing antibody titers against both type A and type B viruses, as well as antibodies that blocked SARS-CoV-2 spike protein binding to the human ACE2 receptor. Moreover, hamsters immunized with the combination vaccine and subsequently challenged with SARS-CoV-2 were protected against weight loss and showed no signs of pneumonia or SARS-CoV-2 replication in the upper or lower respiratory tracts [71].

Some combination vaccines other than qNIV/NVX-CoV2373 are currently in pre-clinical development. For instance, Moderna has recently announced the start of a combination COVID-19/SIV vaccine program [72]. Moreover, nasal spray vaccine candidates produced by Vivaldi Biosciences have been designed to provide protection against both infections and are currently under evaluation in challenge–protection studies in animal models [73].

## 6. Public Acceptance of COVID-19 and Influenza Vaccine Co-Administration and Combined COVID-19/Influenza Vaccines

From the point of view of hesitancy, vaccine co-administration or combination vaccines may encounter additional challenges, owing to misconceptions regarding their efficacy and safety. Specifically, laypeople may believe that that too many vaccines/antigens overload the immune system, may be less effective than the same vaccines administered alone, or may be more reactogenic [74]. Data from the pediatric population, which is the main target for both combination vaccines and vaccine co-administration, suggest that vaccine-hesitant caregivers are less favorable toward combination vaccines and vaccine co-administration [75]. As hesitancy towards both SARS-CoV-2 [76] and SIV [77] is common, in order to design effective targeted health-promotion interventions, it is useful to quantify public acceptance of COVID-19 and SIV co-administration or combination vaccines and its correlates.

In a representative Italian survey conducted in May 2021 [78], 34.1% (95% CI: 32.0–36.2%) of adults expressed “firm willingness” to receive both COVID-19 and SIV simultaneously, while 33.4% (95% CI: 31.3–35.5%) expressed “some willingness” to do so. A subsequent survey conducted in October–November 2021 [79] by the same research group and on an almost identical sample of participants showed a decreasing trend in the acceptance of COVID-19/SIV co-administration: “firm willingness” and “some willingness” were reported by 22.9% (95% CI: 21.3–24.6%) and 36.1% (95% CI: 34.2–38.0%) of Italian adults, respectively. The main significant determinants of positive attitudes towards vaccine co-administration were: compliance with the primary COVID-19 vaccination schedule [adjusted OR (aOR) 7.78]; previous SIV receipt (aOR 1.89); trust in public health institutions (aOR for each 1-to-10 Likert scale point increase: 1.22); male sex (aOR 0.56); younger age (aOR for each 1-year increase: 0.99); willingness to pay for SIV out-of-pocket (aOR 1.79) or to receive a more personalized influenza vaccine (aOR 1.55); perceived influenza severity (aOR 1.36) and recent seeking for influenza-related information (aOR 1.38). The authors concluded that hesitancy towards COVID-19 and SIV co-administration was greater than hesitancy toward the administration of either vaccine alone [79].

Preliminary data [80] from a large University Hospital in Bari (Italy) indicated very high compliance of healthcare workers with COVID-19/SIV co-administration: a total of 60% (1643/2740) received both the third COVID-19 dose and SIV simultaneously. A higher probability of co-administration was seen among physicians (OR 1.93) and residents (OR 2.10), while nurses displayed lower odds (OR 0.75). Moreover, male sex was associated with 43% higher odds of vaccine co-administration, while age was not a significant predictor [80].

Attitudes towards a hypothetical combined COVID-19/SIV vaccine were investigated in a large (*n* = 12,887) United States survey [81]. The acceptance rate of a combined vaccine was 50%, while 45% of respondents stated that they would accept the COVID-19 booster alone, and 58% SIV alone. The highest acceptance (*p* < 0.05) was seen among subjects who received SIV every year (aOR 18.7), identified themselves as democrats (aOR 2.04), were ≥60 years old (aOR 1.37), had a college degree or higher (aOR 1.74), and had an income above the median (aOR 1.29). Conversely, Black/African Americans (aOR 0.60), rural residents (aOR 0.63) and women (aOR 0.65) displayed lower acceptance of a combined COVID-19/SIV vaccine [81]. In Italy, a total of 73.7% (95% CI: 71.7–75.6%) of adults favored receiving a combined COVID-19/SIV vaccine; if such a vaccine were available, 34.8% (95% CI: 32.7–37.0%) said that they would definitely have it, while 35.9% (95% CI: 33.8–38.1%) replied “I think I would have it” [78].

In summary, knowledge, attitudes and practices regarding COVID-19 and SIV co-administration or combination vaccines are components of a complex construct with a bidirectional association: previous SIV receipt is a strong predictor of future COVID-19 vaccination acceptance and vice versa [76,78,79,82,83,84,85]. Moreover, while most individual, structural and contextual determinants of the public’s acceptance of vaccine co-administration or combination vaccines are shared with those of either COVID-19 or SIV administered alone, specific correlates of the former strategies have been identified [79,81].

## 7. Concluding Remarks

The available data [40,41,42] suggest that the co-administration of SIV and COVID-19 vaccines is a feasible option and arouses no safety concerns. Currently, immunological interference between the two vaccines, in terms of the magnitude of the immune response, does not seem to be clinically relevant. Indeed, the principal public health authorities have advocated [44,45,46,47,48,49,50,51,52,53] the strategy of vaccine co-administration. However, it should be borne in mind that the available data on co-administration are mostly based on immunogenicity endpoints with no recognized (for SARS-CoV-2 [58]) or with imperfect (for influenza [86]) correlates of protection. Data on “hard” clinical endpoints, such as efficacy or effectiveness against laboratory-confirmed COVID-19 and influenza, are very few. Given the low level of influenza virus circulation observed since 2020 [4], efficacy or effectiveness estimates of vaccine co-administration or combination vaccine candidates are unlikely to be released in the near future.

While a growing body of evidence suggests that previous SIV receipt may exert non-specific protective effects against some COVID-19-related outcomes (probably thanks to the trained immunity) [22,23,24,27], it is yet to be verified whether the co-administration of SIV and COVID-19 vaccines (or future combination vaccines) may have a synergistic/additive effect on relevant clinical endpoints. There is evidence of an additive effect of co-administering SIV and pneumococcal vaccines [18,87,88]. For instance, in elderly subjects with chronic lung disease, the risk of death was reported to be 34% (95% CI: 6–54%), 70% (95% CI: 57–89%) and 81% (95% CI: 68–88%) lower in subjects who received pneumococcal vaccine alone, SIV alone and both vaccines, respectively, in comparison with non-vaccinated individuals [87].

Laypeople’s attitudes towards COVID-19 and SIV co-administration or combination vaccines are multifaceted and driven by a variety of factors. These determinants should be considered when planning and rolling out future targeted health-promotion interventions to increase immunization uptake. Indeed, the last 2021 winter season in Australia saw a decrease in SIV uptake, which was possibly driven by the exclusion of co-administration strategies [54]. Analogously, policy makers should learn from the 2009 swine influenza pandemic: in the following influenza seasons, in several countries, including Italy [89] and Germany [90], a significant drop in SIV uptake was observed. Lastly, although vaccination is crucial to tackle the burden of both COVID-19 and influenza, non-specific preventive measures such as social distancing, face masks wearing, and hand hygiene will further enhance the vaccine-induced protection.

## Figures and Tables

**Table 1 pharmaceuticals-15-00322-t001:** Rate (%) of solicited local and systemic adverse events in COVID-19 and influenza vaccine co-administration groups, as compared with groups to whom either vaccine was administered alone.

Adverse Event	Comparison		Vaccine Administration Pattern			Reference
	COVID-19 Vaccine	SIV	COVID-19 + SIV	COVID-19 Alone	SIV Alone	
Any local, %	ChAdOx1 ^1^	QIVc ^1^	84	81	–	[40]
	BNT162b2 ^1^	QIVc ^1^	96	94	–	[40]
	ChAdOx1 ^1^	QIVr ^1^	85	86	–	[40]
	BNT162b2 ^1^	QIVr ^1^	96	89	–	[40]
	ChAdOx1 ^2^	aTIV ^2^	77	65	–	[40]
	BNT162b2 ^2^	aTIV ^2^	76	79	–	[40]
	mRNA-1273 ^2^	hdQIV ^2^	86	91	62	[42]
	NVX-CoV2373 ^1^	QIVc ^1^	73	63	39	[41]
	NVX-CoV2373 ^2^	aTIV ^2^	39	35	46	[41]
Any systemic, %	ChAdOx1 ^1^	QIVc ^1^	81	83	–	[40]
	BNT162b2 ^1^	QIVc ^1^	87	81	–	[40]
	ChAdOx1 ^1^	QIVr ^1^	74	72	–	[40]
	BNT162b2 ^1^	QIVr ^1^	89	82	–	[40]
	ChAdOx1 ^2^	aTIV ^2^	72	62	–	[40]
	BNT162b2 ^2^	aTIV ^2^	59	71	–	[40]
	mRNA-1273 ^2^	hdQIV ^2^	80	84	49	[42]
	NVX-CoV2373 ^1^	QIVc ^1^	62	50	47	[41]
	NVX-CoV2373 ^2^	aTIV ^2^	39	28	55	[41]

^1^ Working-age adults (18–64 years); ^2^ older adults (≥65 years); aTIV, MF59-adjuvanted trivalent influenza vaccine; hdQIV, high-dose quadrivalent influenza vaccine; QIVc, cell-based quadrivalent influenza vaccine; QIVr, recombinant quadrivalent influenza vaccine.

**Table 2 pharmaceuticals-15-00322-t002:** Hemagglutination inhibition IgG geometric mean ratios against influenza vaccine strains in COVID-19 + seasonal influenza vaccine co-administration groups, as compared with groups to whom either vaccine was administered alone.

Influenza Vaccine	Strain	COVID-19 Vaccine [Reference]			
		BNT162b2 [40]	mRNA-1273 [42]	ChAdOx1 [40]	NVX-CoV2373 [41]
QIVc ^1^	A(H1N1)pdm09	1.05 (0.91–1.21) ^4^	–	1.05 (0.91–1.21) ^4^	1.09 (ns) ^6^
	A(H3N2)	1.06 (0.95–1.18) ^4^	–	1.08 (0.96–1.21) ^4^	1.08 (ns) ^6^
	B/Victoria	1.03 (0.93–1.14) ^4^	–	1.05 (0.94–1.18) ^4^	1.03 (ns) ^6^
	B/Yamagata	0.94 (0.85–1.05) ^4^	–	0.98 (0.88–1.10) ^4^	0.99 (ns) ^6^
QIVr ^1^	A(H1N1)pdm09	1.38 (1.11–1.71) ^4^	–	0.86 (0.74–0.99) ^4^	–
	A(H3N2)	1.03 (0.87–1.23) ^4^	–	1.03 (0.91–1.15) ^4^	–
	B/Victoria	1.20 (1.02–1.42) ^4^	–	1.07 (0.96–1.19) ^4^	–
	B/Yamagata	1.24 (1.05–1.47) ^4^	–	1.06 (0.94–1.18) ^4^	–
aTIV ^2^	A(H1N1)pdm09	1.05 (0.87–1.27) ^4^	–	1.15 (1.01–1.32) ^4^	1.41 (ns) ^6^
	A(H3N2)	1.18 (1.02–1.37) ^4^	–	1.00 (0.89–1.11) ^4^	0.87 (ns) ^6^
	B/Victoria	1.08 (0.94–1.25) ^4^	–	1.01 (0.91–1.12) ^4^	0.54 (ns) ^6^
	B/Yamagata ^3^	1.00 (0.86–1.15) ^4^	–	0.92 (0.83–1.03) ^4^	0.81 (ns) ^6^
hdQIV ^2^	A(H1N1)pdm09	–	0.99 (ns) ^5^	–	–
	A(H3N2)	–	0.91 (ns) ^5^	–	–
	B/Victoria	–	0.97 (ns) ^5^	–	–
	B/Yamagata	–	0.91 (ns) ^5^	–	–

^1^ Working-age adults (18–64 years); ^2^ older adults (≥65 years); ^3^ strain was not present in the vaccine formulation; ^4^ seasonal influenza vaccine first vs. placebo first; ^5^ mRNA-1273 + hdQIV vs. hdQIV alone, geometric mean ratios were calculated from the geometric mean titers provided by the authors; ^6^ NVX-CoV2373 + seasonal influenza vaccine vs. placebo + seasonal influenza vaccine, geometric mean ratios were calculated from the geometric mean titers provided by the authors; aTIV, MF59-adjuvanted trivalent influenza vaccine; hdQIV, high-dose quadrivalent influenza vaccine; ns, non-significant (*p* > 0.05); QIVc, cell-based quadrivalent influenza vaccine; QIVr, recombinant quadrivalent influenza vaccine.

**Table 3 pharmaceuticals-15-00322-t003:** Anti-spike IgG geometric mean ratios in COVID-19/influenza vaccine co-administration groups, as compared with groups to whom either vaccine was administered alone.

Influenza Vaccine	COVID-19 Vaccine [Reference]			
	BNT162b2 [40]	mRNA-1273 [42]	ChAdOx1 [40]	NVX-CoV2373 [41]
QIVc ^1^	0.90 (0.80–1.01) ^3^	–	0.92 (0.81–1.04) ^3^	0.66 (NA) ^5^
QIVr ^1^	0.86 (0.72–1.03) ^3^	–	0.92 (0.81–1.04) ^3^	–
aTIV ^2^	0.97 (0.83–1.13) ^3^	–	1.02 (0.91–1.14) ^3^	0.71 (NA) ^5^
hdQIV ^2^	–	0.97 (0.79–1.19) ^4^	–	–

^1^ Working-age adults (18–64 years); ^2^ older adults (≥65 years); ^3^ COVID-19 vaccine + placebo vs. COVID-19 vaccine + seasonal influenza vaccine; ^4^ mRNA-1273 + hdQIV vs. hdQIV alone; ^5^ NVX-CoV2373 + seasonal influenza vaccine vs. placebo + NVX-CoV2373 alone, geometric mean ratios were calculated from the geometric mean titers provided by the authors; aTIV, MF59-adjuvanted trivalent influenza vaccine; hdQIV, high-dose quadrivalent influenza vaccine; NA, not available; QIVc, cell-based quadrivalent influenza vaccine; QIVr, recombinant quadrivalent influenza vaccine.

## Data Availability

Not applicable.

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
