# Peer review of "COVID-19 and Seasonal Influenza Vaccination: Cross-Protection, Co-Administration, Combination Vaccines, and Hesitancy"

_pharmaceuticals, 2022, doi:10.3390/ph15030322_

Round 1

Reviewer 1 Report

The manuscript entitled “COVID-19 and Seasonal Influenza Vaccination: From Co-Administration Through Hesitancy to Combination Vaccines”, is well written and interesting, offering important contributions of consistent scientific data exclusively restricted to the reality of populations on the European continent.

I suggest that the authors add a brief paragraph in the conclusion topic highlighting that although vaccines are the main strategy to reduce the chances of severe cases and even death from viruses such as COVID-19 and others cited, asymptomatic infection or mild symptoms are still possible, and that other associated secondary prevention strategies are necessary such as physical distancing, use of masks, adequate nutrition, ignoring political groups against vaccination, etc...

Author Response

Comment: The manuscript entitled “COVID-19 and Seasonal Influenza Vaccination: From Co-Administration Through Hesitancy to Combination Vaccines”, is well written and interesting, offering important contributions of consistent scientific data exclusively restricted to the reality of populations on the European continent.

Reply: Thank you for your interest in our paper. All your comments have been addressed.

Comment: I suggest that the authors add a brief paragraph in the conclusion topic highlighting that although vaccines are the main strategy to reduce the chances of severe cases and even death from viruses such as COVID-19 and others cited, asymptomatic infection or mild symptoms are still possible, and that other associated secondary prevention strategies are necessary such as physical distancing, use of masks, adequate nutrition, ignoring political groups against vaccination, etc...

Reply: As suggested, we have now clearly stated the importance of non-specific interventions to reduce the burden of both COVID-19 and influenza.

Reviewer 2 Report

The review by Domnich et al., adequately describes our current understanding of CoVID-19 and influenza vaccination. This area of research is of interest to many stakeholders. The manuscript is generally well written but there are a small number of edits that would improve it.

Non-essential words make the title a little cumbersome. I would recommend "COVID-19 and Seasonal Influenza Vaccination: Cross-Protection, Co-Administration and Combination Vaccines". The word "acceptance" on line 16 could be replaced with "hesitancy" if the authors feel that hesitancy needs to be included in the abstract or title.

line 86, the I2 percentage from the random effects model does not belong in the paper and will not be relevant for most readers.

line 151, the sentence would be better if "what concerns" is removed.

line 190, ns is included in the footnotes for table 1 but isn't used in the table. The main text indicates that some (all?) of the differences are non-significant and this should be indicated in the relevant lines of the table.

Tables 2 and 3 should have [Reference] rather than [Ref].

line 258, the immunological interactions don't occur between the antigens. I recommend altering the text to "...which may be due to, not only the combination of antigens, but also other vaccine components..."

line 336, "believe" rather than "belief"

line 390, "interference" rather than "inference"

lines 406-8, I would list the pneumococcal vaccine alone risk first to better illustrate the additive effect, but the change isn't necessary.

line 510, a hyphen is missing (H1N1-mediated) in the title.

Author Response

Comment: The review by Domnich et al., adequately describes our current understanding of CoVID-19 and influenza vaccination. This area of research is of interest to many stakeholders. The manuscript is generally well written but there are a small number of edits that would improve it.

Reply: Thank you for your interest in our paper. All your comments have been addressed.

Comment: Non-essential words make the title a little cumbersome. I would recommend "COVID-19 and Seasonal Influenza Vaccination: Cross-Protection, Co-Administration and Combination Vaccines". The word "acceptance" on line 16 could be replaced with "hesitancy" if the authors feel that hesitancy needs to be included in the abstract or title.

Reply: As suggested, the title has been modified.

Comment: line 86, the I2 percentage from the random effects model does not belong in the paper and will not be relevant for most readers.

Reply: The I2 value has been deleted.

Comment: line 151, the sentence would be better if "what concerns" is removed.

Reply: The phrase has been modified.

Comment: line 190, ns is included in the footnotes for table 1 but isn't used in the table. The main text indicates that some (all?) of the differences are non-significant and this should be indicated in the relevant lines of the table.

Reply: This has been corrected.

Comment: Tables 2 and 3 should have [Reference] rather than [Ref].

Reply: This has been corrected.

Comment: line 258, the immunological interactions don't occur between the antigens. I recommend altering the text to "...which may be due to, not only the combination of antigens, but also other vaccine components..."

Reply: The phrase has been amended.

Comment: line 336, "believe" rather than "belief"

Reply: This has been corrected.

Comment: line 390, "interference" rather than "inference"

Reply: This has been corrected.

Comment: lines 406-8, I would list the pneumococcal vaccine alone risk first to better illustrate the additive effect, but the change isn't necessary.

Reply: The phrase has been amended.

Comment: line 510, a hyphen is missing (H1N1-mediated) in the title.

Reply: This has been corrected.